# Stem Cell Theory of Cancer: Implications for Drug Resistance and Chemosensitivity in Cancer Care

**DOI:** 10.3390/cancers14061548

**Published:** 2022-03-18

**Authors:** Shi-Ming Tu, Charles C. Guo, Diana S. -L. Chow, Niki M. Zacharias

**Affiliations:** 1Division of Hematology/Oncology, Winthrop P. Rockefeller Cancer Institute, University of Arkansas for Medical Sciences, Little Rock, AR 72205, USA; 2Department of Pathology, The University of Texas MD Anderson Cancer Center, Houston, TX 77030, USA; ccguo@mdanderson.org; 3Department of Pharmacological and Pharmaceutical Sciences, University of Houston College of Pharmacy, Houston, TX 77030, USA; dchow@uh.edu; 4Department of Urology, The University of Texas MD Anderson Cancer Center, Houston, TX 77030, USA; nmzacharias@mdanderson.org

**Keywords:** drug resistance, cancer stem cells, unified theory, clonal origin, chemosensitivity, chronotherapy, circadian rhythm, drug development, therapy development

## Abstract

**Simple Summary:**

Science and history teach us that stemness properties pave all drug resistance pathways. Evidence and experience inform us that stemness origin and nature etch all cancer hallmarks. A stem cell origin of drug resistance encompasses heterogeneity and dormancy, embraces ABC transporters and DNA repairs, and explicates chemotherapy and chronotherapy. It alludes to a unified theory of cancer and suggests that cancer is a stem cell disease—uniting chemoresistance with chemosensitivity, connecting progenitor cells with progeny cells, and linking multicellularity with the microenvironment. Importantly, it clarifies genetic content vs. cellular context, delineates drug vs. therapy development, and enlightens precision medicine vs. integrated medicine and targeted therapy vs. multimodal therapy in cancer care.

**Abstract:**

When it concerns cancer care and cancer therapy, drug resistance is more than an obstacle to successful treatment; it is a major cause of frustration in our attempts to optimize drug development versus therapy development. Importantly, overcoming the challenges of drug resistance may provide invaluable clues about the origin and nature of cancer. From this perspective, we discuss how chemoresistance and chemosensitivity in cancer therapy could be directly linked to the stem cell origin of cancer. A stem cell theory of cancer stipulates that both normal stem cells and cancer stem cells are similarly endowed with robust efflux pumps, potent antiapoptotic mechanisms, redundant DNA repair systems, and abundant antioxidation reserves. Cancer stem cells, like their normal stem cell counterparts, are equipped with the same drug resistance phenotypes (e.g., ABC transporters, anti-apoptotic pathways, and DNA repair mechanisms). Drug resistance, like other cancer hallmarks (e.g., tumor heterogeneity and cancer dormancy), could be intrinsically ingrained and innately embedded within malignancy. We elaborate that cellular context and the microenvironment may attenuate the effects of cancer treatments. We examine the role of circadian rhythms and the value of chronotherapy to maximize efficacy and minimize toxicity. We propose that a stem cell theory of drug resistance and drug sensitivity will ultimately empower us to enhance drug development and enable us to improve therapy development in patient care.

## 1. Introduction

“*The best doctor gives the least medicines*.”—Benjamin Franklin.

Novel targeted drugs, such as trastuzumab, imatinib, and abiraterone, have revolutionized cancer care. However, even with these new drugs, resistance is still an issue. Drug resistance is more than an obstacle to successful treatment; it is a major cause of frustration in our attempts to achieve optimal clinical outcomes for patients with cancer.

Drug resistance is an evolutionary process. Bacteria and cancers that are initially responsive to antibiotic and chemotherapy, respectively, over time will become unresponsive. However, because cancer may be drug resistant without ever being exposed to or treated with certain drugs, it is conceivable that drug resistance could be in some instances an intrinsic property in the very origin and constitution of cancer.

In other words, drug resistance could be innately embedded within malignancy like other cancer hallmarks, such as metastasis and heterogeneity, because of its stemness origin and nature [1]. A stem cell theory of cancer postulates that cancer is endowed with robust efflux pumps, potent anti-apoptotic mechanisms, redundant DNA repair systems, abundant antioxidation reserves, and the like. It stipulates that we stringently protect and preserve our normal stem cells (the fountain of all cells and the spring of our very being) by all means, but unwittingly embolden and invigorate cancer stem cells in the same process (Figure 1).

From this perspective, we present a brief history of drug resistance in cancer care. We discuss how chemoresistance and chemosensitivity in cancer therapy could be directly linked to the stem cell theory of cancer. We elaborate that a stem cell origin of heterogeneity and dormancy may attenuate the effects of cancer treatments. We investigate the role of circadian rhythms and the value of chronotherapy to maximize efficacy and minimize toxicity. We propose that a stem cell theory of drug resistance and drug sensitivity will ultimately empower us to enhance drug development and enable us to improve therapy development in patient care. We become better physicians when we know why and how to give less medicine.

## 2. Brief History

In 1882, Koch established an infectious cause of tuberculosis [2]. However, it was not until 1944 when Waksman first isolated an effective anti-tuberculosis drug, streptomycin [3]. The development of isoniazid in 1951 and rifampin in 1957 revolutionized the treatment of tuberculosis. Unfortunately, drug resistance occurred soon after the administration of these drugs. Similarly, resistance to chloroquine for the treatment of malaria (the early 1960s) and zidovudine for the treatment of human immunodeficiency virus (the late 1980s) was also prevalent, if not inevitable.

In 1943, Gilman and Goodman used nitrogen mustard to treat lymphoma [4]. In 1948, Farber used antifolate to treat acute lymphoblastic leukemia [5]. Although they demonstrated clinical response and symptomatic improvement, drug resistance was indomitable and immediate (occurred within weeks).

In 1987, Thiebaut et al. discovered P-glycoprotein, a multidrug resistance (MDR) membrane transport protein encoded by ABCB1 (MDR1) [6]. In 2000, Schmitt et al. demonstrated that the overexpression of the anti-apoptotic protein Bcl-2 produced an MDR phenotype in primary lymphomas in vivo [7]. Intriguingly, this effect was dramatically reduced when the primary lymphomas were subjected to long-term culture and completely absent in the standard clonogenic assay.

An important lesson learned from these studies was to give combined non-cross-reactant drugs, rather than a single drug by itself, to delay, if not to prevent, the emergence of drug-resistant strains and variants. Nevertheless, Schmitt et al.’s study [7] suggested that when drug resistance occurred in a cancer, as opposed to a virus, bacterium, or protozoa, it was more than a genetic event—it involved intricate multicellular processes.

Another lesson learned was that all mechanisms that are involved in and/or contribute to drug resistance in cancer—transporter proteins, anti-apoptosis, DNA repair, cell cycle arrest/autophagy, differentiation/heterogeneity, et cetera—somehow implicate stemness properties and invoke stem-like pathways [8,9,10,11,12,13,14,15,16,17].

When myriad disparate mechanisms of drug resistance are in play and may be linked through stemness, history teaches us about a stem cell origin of cancer, tells us that cancer is a stem cell disease, and alludes to a unified theory of cancer [18,19].

## 3. Chemotherapy

Chemotherapy is often maligned, because it entails horrendous toxic effects and inherent drug resistance. Nonetheless, one should not forget that it is also one of the very few treatments that have cured certain cancers [20].

Perhaps we should not disparage chemotherapy, when it is quite capable of curing even advanced metastatic cancers. When surgery or radiation therapy is curative, it is usually within the context of an early-stage tumor that is still confined or contained. Although we think that immunotherapy (and hope that targeted therapy) may be able to cure some advanced metastatic cancers, the fact is that most cancers invariably become resistant and recurrent with these novel treatments, just like they do with traditional chemotherapy.

Importantly, neither immunotherapy nor targeted therapy are free from horrendous toxic effects and inherent drug resistance. Often enough, the side effects and complications from these innovative treatments can be just as debilitating and devastating as those caused by traditional chemotherapies. An unspoken truth is that most cancers will become resistant to these appealing and promising treatment modalities, too. Perhaps out of exasperation (or desperation), we are combining these supposedly precise immunotherapies and targeted therapies with notoriously imprecise chemotherapies.

Instead of avoiding chemotherapy, we seem to be drawing toward it. There must be an overriding and underlying reason why chemotherapy is often effective and sometimes curative. Understandably, when a treatment is curative, it is unethical and irresponsible not to use it. The dilemma is that we do not fully understand why chemotherapy is effective. The challenge is that we need to know how to harness it at the right time and in the proper manner to maximize benefits and minimize risks. To accomplish this goal, we need a better understanding of the origin and nature of cancer and how chemotherapy may exploit cancer’s unique idiosyncrasies and putative vulnerabilities.

## 4. Chemoresistance

Before we explore chemoresistance, let us discuss chemosensitivity. To understand the cases in which chemotherapy may not be effective, it is invaluable to know why and how chemotherapy works in the first place. It will be easier to pinpoint the misstep or trace the missing cog when we have a clearer idea about the entire pathway or the whole gearwheel.

Savage connected chemosensitivity to proapoptotic pressures during development [21]. He pointed out that a gestational trophoblast tumor arises during nuclear fusion (after fertilization) and is 100% curable with a single dose of methotrexate. However, when the tumor forms after nuclear fusion (i.e., gestational choriocarcinoma), the cure rate decreases to about 50% with three cycles of combined bleomycin, etoposide, and cisplatin.

Childhood malignancies that arise during gastrulation (when the single-layer blastocyst segregates into the three germ layers of ectoderm, mesoderm, and endoderm) are also particularly sensitive to chemotherapy due to their shortened cell cycle without cell cycle arrest and robust DNA repair in response to DNA damage. Hence, the cure rate of neuroblastoma with chemotherapy is up to 90%, and of Wilms tumor, 80%.

Before puberty, an ovum has already commenced meiosis and is suspended in mid-meiosis, whereas a spermatogonium has not yet committed to meiosis but is poised to start differentiation and spermatogenesis. The overall cure rate of germ cell tumor of the testis (TGCT) with chemotherapy is about 90%. Intriguingly, prepubertal and postpubertal ovarian teratoma tends to be less malignant compared with postpubertal testicular teratoma, even though all teratomas are completely chemoresistant [20].

In addition, B cell acute lymphocytic leukemia arises from pro-B cells in which VDJ rearrangement (RAG1 and RAG2) is in full bloom, and the cure rate in children with chemotherapy is about 90%. Furthermore, diffuse large B cell lymphoma, Hodgkin’s lymphoma, and Burkitt’s lymphoma arise from germinal center B cells, in which somatic hypermutation (cytidine deaminase) is full blown, and the cure rate with chemotherapy is around 80%.

## 5. Stem Cell Origin

One way to elucidate chemoresistance vs. chemosensitivity is to investigate their mechanisms of action in the context of a stem cell theory of cancer. When drug resistance involves stem-like properties or pathways in both normal and cancer stem cells, it is consistent with a stem cell origin of drug resistance [8,9,10,11,12,13,14,15,16,17]. When cancer stem cells (CSCs) are innately endowed with MDR capabilities in the form of ATP-binding cassette (ABC) transporters, anti-apoptotic mechanisms, glutathione-S-transferase (GST) expressions, they are inherently resistant to many standard and novel therapies and are naturally predisposed to tumor regrowth and relapse.

Specifically, ABC transporters play an important role protecting both normal and cancer stem cells from noxious toxins and potent drugs [8,9,10,11]. Sun et al. found that stem-like CD44^+^CD24^−/low^ cells isolated from several breast cancer cell lines, including MDA-MB-231, expressed higher levels of ABCG2 than non-stem cells [9]. Sims-Mourtada et al. showed that stemness signaling (through Hh) regulated ABCB1 and ABCG2, leading to therapy-resistant CSCs [11].

Similarly, the same intrinsic and extrinsic apoptotic pathways, the TRADD/NFkB survival pathway and the growth factor receptors PI3K/AKT pro-survival signaling axis, enable both normal and cancer stem cells to survive and thrive under duress when exposed to or treated with chemicals during embryogenesis and carcinogenesis [12,13]. 

Likewise, what is considered good (e.g., the protection and preservation of our DNA and other vital cellular constituents) in a normal stem cell (i.e., detoxification and antioxidation) is bad in a cancer stem cell (i.e., drug resistance). Hence, CSCs have higher GST levels than non-CSCs [15,16,17]. Again, it appears that GST mediates drug resistance through various stemness pathways and networks, including those involving MAP kinase [17] and NFkB [22].

It is uncanny that the same processes of MDR may be indispensable in a normal stem cell (e.g., for purposes of protection and survival) but intractable in a cancer stem cell (i.e., drug resistance). In theory, the interplay among ABC transporters, anti-apoptotic pathways, and GST levels could be synergistic or antagonistic and death defining or death defying, depending on whether it occurs in a CSC or a non-CSC. In practice, whether the operation of ABC transporters, the modulation of anti-apoptotic pathways, and the expression of GST activates or attenuates drug resistance depends on the cell type or context [14,19].

## 6. Cellular Context

It is evident that all cancer hallmarks, including drug resistance, have a stemness earmark, e.g., pluripotency [23], epithelial-to-mesenchymal transition (EMT) [24], autophagy [14], et cetera. When we consider stemness, the epigenome trumps the genome. Cellular context (which includes cell–cell interactions and the microenvironment) vetoes genetic defects.

Contrary to conventional wisdom, Flinders et al. demonstrated that epigenetic changes rather than genetic mutations engendered drug resistance [23]. The same genome elicits different inborn clonal outputs that affect drug resistance [25]. Conversely, Turati et al. found that chemotherapy exerted little impact on genetic heterogeneity [26]. Up to 40% of recurrent resistant tumors displayed no discernible new genetic defects [27].

In many respects, TGCT provides a stark testimony to the relevance of cellular context vs. genetic signature in drug resistance. After all, TGCT is a prototype stem cell tumor with the ability to differentiate into many lineages and form mixed tumors with varied components, some of which (e.g., embryonal carcinoma) are exquisitely chemosensitive, while others can be completely chemoresistant (e.g., teratoma). Importantly, both embryonal carcinoma and teratoma in a mixed non-seminomatous germ-cell tumor contain a similar, if not the same, genetic makeup (e.g., i[12p]) due to their common clonal origin [28,29,30].

Therefore, one of the most well-differentiated tumors, namely mature teratoma, is also one of the most drug-resistant tumors. When we understand cellular context rather than presume genetic mutation in a mixed TGCT, we cure it with integrated rather than precision medicine and overcome drug resistance with multimodal rather than targeted therapy. In other words, we remove the residual teratoma with surgery after chemotherapy rather than administering more chemotherapy.

Because stemness begets heterogeneity and heterogeneity belies simplicity, a stem cell theory of cancer predicts that the epigenetic vs. genetic expression of various CSCs and non-CSCs within the same tumor may be another source of drug resistance. When we rely on precision medicine and design targeted therapy to treat the parts rather than the whole tumor, and when we tackle genetic mutation without proper regard for cellular context, we may be messing with drug resistance and missing the cancer target in drug development for cancer care.

## 7. DNA Repair

Human stem cells possess a highly efficient DNA repair network that becomes less efficient upon differentiation. In addition, many adult stem cells are quiescent, remaining at the G0 phase of the cell cycle, which minimizes the chance of replication error.

When DNA damage is not repaired, stem cells easily undergo senescence, cell death, or differentiation as part of their response to DNA damage in order to avoid the propagation of genetic mutations and genomic alterations to their offspring.

Notably, there is higher expression of mismatch repair proteins (MSH2, MSH6, MLH1, and PMS2), base excision repair proteins (AAG and APEX), and MGMT in CD34^+^ stem cells compared with the terminally differentiated CD34^−^ cells [31].

Furthermore, the rates of the removal of DNA adducts, the resealing of repair gaps, and the resistance to DNA-reactive drugs were higher in stem (CD34^+^ 38^−^) than in mature (CD34^−^) or progenitor (CD34^+^ 38^+^) cells from the same individual [32].

Not surprisingly, embryonic stem (ES) cells have the highest DNA repair capacity, yet they have an abbreviated cell cycle (G1, S, G2, and M) with a short G1 and a quick transition from G1 to S due to increased CDK4 and cyclin D2.

The unique G1 kinetics and partial deficiency in the G1/S checkpoint in ES cells allow damaged cells to progress into S, in which the DNA damage is amplified, leading to cell death (hence the chemosensitivity of embryonal carcinoma, which is derived from an ES-like cell). Premature differentiation (with the induction of cell cycle arrest and impaired DNA repair) and senescence (stalled or dead-end cells) are alternative outcomes.

In cases of severe or excessive DNA damage, p53 induces the apoptosis or senescence of those cells afflicted. The activation of p53 also suppresses pluripotency genes, such as Nanog, allowing differentiation to proceed.

## 8. Asymmetric Division

Another clue hinting at a stem cell origin of drug resistance and sensitivity can be discerned from the discovery of microsatellite instability (MSI) and its link to aneuploid vs. diploid tumors. MSI is a condition of genetic hypermutability that results from impaired DNA mismatch repair.

For example, there are two types of colorectal cancers (CRCs). One is non-hypermutated and chromosomally unstable (i.e., aneuploid), while the other is hypermutated with MSI (and tends to be diploid) [33].

When DNA damage is not properly repaired in stem cells, mistakes in asymmetric division may occur, aggravating genetic instability and resulting in aneuploidy.

Therefore, MSI tumors are likely to be derived from defective “differentiated” progeny cells (rather than progenitor stem cells) in which asymmetric division and aneuploidy do not occur [34]. Consequently, MSI is associated with a better overall prognosis in CRC [35,36].

A stem cell origin of cancer conceives a hierarchical order of stem cells with a continuum of stemness and differentiation and a vast spectrum of DNA repair capacity and apoptotic capability. It also envisions diverse lineages of progeny differentiated cells derived from those progenitor stem cells equipped with various life spans, antioxidation reserves, genetic signatures, and neo-antigen profiles.

In general, early progenitor stem cells are proficient in apoptotic, senescent, and differentiation mechanisms, which render them relatively chemosensitive. Interestingly, some late progeny mature cells are also therapeutically sensitive to a variety of treatment modalities due to their proapoptotic capabilities, low antioxidation reserves, and hypermutability/immunogenicity portfolios.

## 9. A Fortuitous Experiment

According to the scientific method, we design experiments to test hypotheses, not to generate them. Otherwise, the results from the experiment can be misleading and confounding.

Hence, when it concerns an experiment to test drug resistance, it may not be easy to discern the prognostic (nature of disease) or predictive (efficacy of drug) value of a drug when we do not have a proper hypothesis about the origin and nature of drug resistance.

It turns out that induced pluripotent stem cells (iPSCs) may be a fortuitous experimental model to elucidate drug resistance in stem cells vs. differentiated cells. After all, iPSCs are supposed to be derived from differentiated cells but are reprogrammed with stemness genes, i.e., they are artificial stem cells with mature cell origins.

Gore et al. showed that at least half of the protein-coding point mutations present in 22 iPSC cell lines were acquired during the reprogramming process, independent of the reprogramming method [37].

Importantly, the iPSC cell line that displayed a normal karyotype (i.e., diploid) harbored MSI with reduced DNA repair capacities in three out of four DNA repair pathways. In other words, reprogramming in iPSCs is incomplete [38]—iPSCs are still progeny rather than progenitor stem cells.

It is of interest that iPSCs are predisposed to malignant formation [39,40,41]; perhaps the stemness genes made it so. However, it is plausible that the same aberrant genes in a different cellular context, e.g., ESCs vs. iPSCs, may engender different malignancies with different tumor phenotypes (if not genotypes), including drug resistance, with profound prognostic (and diagnostic) rather than predictive implications.

## 10. Chronotherapy

In many respects, drug resistance is closely tied to the efficacy of treatment. When a treatment is not effective or less effective, drug resistance may be responsible in one way or another.

An unappreciated aspect of drug resistance is time and timing. We may be cognizant that the circadian cycle affects our health by regulating our sleep, metabolism, and hormones [42,43]. We may be aware that genes and a diurnal schedule influence the physiology and pharmacology of drug resistance vs. sensitivity, as well as the efficacy and toxicity of cancer therapy [44,45]. However, we may not realize that the cellular origin and type of cell—a progenitor vs. a progeny cell—in which the genes reside and the clock clicks also matter.

Therefore, when it concerns the function and activity of a gene, we may need to consider its temporal context, too. For example, the activity of dihydropyrimidine dehydrogenase (DPD) in human mononuclear cells increases by 40% around midnight. Because DPD catabolizes 5-fluorouracil within cells, patients experience the improved tolerability of this drug between midnight and 4:00 am [44]. Similarly, Qian et al. showed that the timing of immune checkpoint inhibitor infusions impacted the survival of patients with advanced melanoma [45]. There is a diurnal pattern due to the circadian rhythm with respect to our adaptive immune response—more sensitive (i.e., less resistant) when we stimulate our immune system during the daytime vs. the evening to counter cancer.

Therefore, when it concerns the function and activity of a gene, we should not forget or ignore its cellular or temporal context. Although the gene is pivotal, its function and activity in a progenitor stem-like cell vs. progeny differentiated cell may be paramount. After all, cellular context is interconnected with temporal context. When a progenitor cell differentiates into a progeny cell, the circadian clock changes [46,47,48], i.e., when the cell is rewired, the clock is reset. This is understandable, given that the molecular clock governs stem cell maintenance and organ physiology and dictates tissue homeostasis and regeneration.

Remarkably, cancer chronotherapy could be another aspect of drug resistance or sensitivity that is intrinsically and intimately associated with a stem cell origin of cancer. When we think of drug vs. therapy development, we certainly need to think about the genome, transcriptome, metabolome, et cetera. Perhaps we also need to consider a “stemnome,” if not a “chrononome,” in a complete equation to solve the problem of drug resistance in cancer care.

## 11. Drug Development

Nowadays, we hail drug development. The discovery of an effective drug is lucrative and rewarding. However, the design of innovative drugs is expensive and risky. When most novel drugs fail and only a few provide incremental or marginal clinical benefits, something seems amiss with the promises and basic premises of drug development. Perhaps drug resistance is to blame. Perhaps our main idea about the origin and nature of cancer is at fault.

Perhaps we have neglected therapy development in our zeal to promote drug development. Unquestionably, we need drugs to treat cancer, but how and when to use those drugs, in the right patients and for the right cancer types, become key. Given the complex and dynamic nature of cancer, we may need to provide integrated rather than precision medicine and prescribe multimodal rather than targeted therapy to maximize clinical outcome and optimize patient care. The art of healing still beats the science of medicating.

One way to defeat drug resistance is to develop better drugs. Another way is to develop better treatments with better ideas (using the same or even fewer drugs). For example, the cure rate of TGCT has steadily increased from about 70% (in the 1970s) to over 90% (currently) without the development of any new or special drugs [20]. Therefore, TGCT is highly curable because we have applied the principles of therapy development rather than the acclaims of drug development: treat the chemosensitive, systemic cancer (e.g., embryonal carcinoma) with chemotherapy in the right way at the right time, and remove the residual localized chemoresistant tumor (e.g., teratoma) with surgery after chemotherapy [30,49]. Chemotherapy or surgery alone is inadequate to cure TGCT. Targeting the same genetic defects (such as i[12p]) in a mixed TGCT is likely to be moot, if not futile. At best, any clinical benefit will be marginal and transient, and the treatment may be effective for the wrong reasons.

Although drug development (such as with enzalutamide, apalutamide, and darolutamide) has provided us effective drugs in the care of advanced castration-resistant prostate cancers, they are not curative treatments (Table 1, row 3). In a malignancy in which we invariably attain gratifying responses with androgen deprivation therapy (ADT), we also need therapy development above and beyond drug development to achieve durable remissions. This requires that we know and understand the origin and nature of drug resistance in relation to progenitor stem-like cells (e.g., AR^−^, ADT resistant, and non–PSA producing) and progeny differentiated cells (AR^+^, ADT sensitive, and PSA producing). An effective maintenance regimen to control the former cell types may not be clinically beneficial in an upfront setting when we also need to treat the bulk of the latter cell types [50]. Integrating treatments that target both tumor components is more likely to improve the overall clinical outcome than treating either alone.

## 12. Conclusions

For a cancer doctor, drug resistance is a reality and seems inevitable. For a cancer researcher, it is a challenge and seems inexplicable.

History teaches us that all drug resistance pathways lead to stemness properties. Science informs us that all cancer hallmarks have a stemness origin and nature. A stem cell origin of drug resistance encompasses heterogeneity and dormancy, embraces ABC transporters and DNA repairs, and explicates chemotherapy and chronotherapy. It alludes to a unified theory of cancer and suggests that cancer is a stem cell disease.

A stem cell theory of cancer unites chemoresistance with chemosensitivity, connects progenitor cells with progeny cells, and links multicellularity with the microenvironment. Importantly, it clarifies genetic content vs. cellular context, delineates drug vs. therapy development, and enlightens precision medicine vs. integrated medicine and targeted therapy vs. multimodal therapy in patient care (Figure 2).

Undoubtedly, when we understand the origin of cancer and of drug resistance in cancer, and when we know why and how to give less medicine, we become better physicians and healers of patients with cancer.

## Figures and Tables

**Figure 1 cancers-14-01548-f001:**
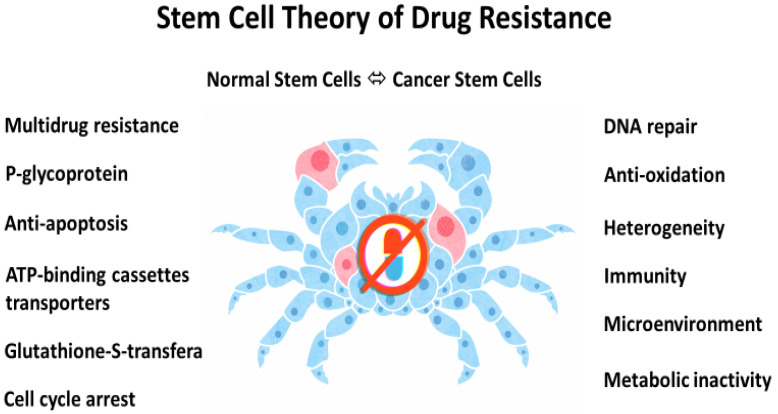
Stem cell origin of drug resistance in cancer. According to Greek mythology, Karkinos (a crab) is the symbol of cancer. Illustration by Benjamin Tu.

**Figure 2 cancers-14-01548-f002:**
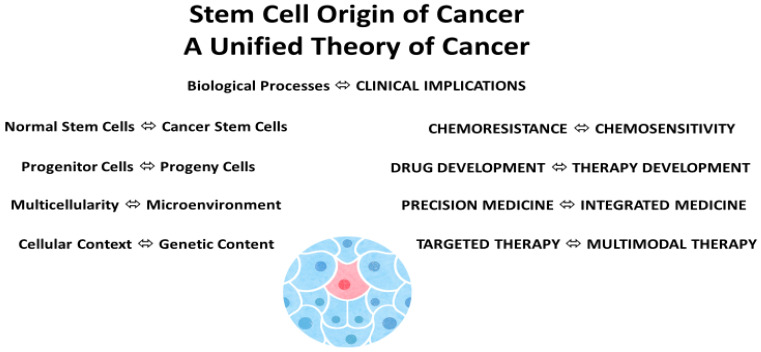
Stem cell origin and a unified theory of cancer: biological processes and clinical implications. Illustration by Benjamin Tu.

**Table 1 cancers-14-01548-t001:** Drug development in cancer care: an abbreviated and representative version.

Target	Drugs (Date First Approved by FDA)	Cancer Type
Multiple/CD30	Mechlorethamine * (1949), vincristine (1963), vinblastine (1965), procarbazine (1969), bleomycin (1973), doxorubicin (1974), dacarbazine (1975), brentuximab (2018)	Hodgkin lymphoma ^1,2^
Multiple	Vinblastine (1965), bleomycin (1973), cisplatin (1978), etoposide (1983), ifosfamide (1988), paclitaxel (1992)	Testis ^3–5^
AR	Flutamide (1989), bicalutamide (2008), abiraterone (2011), enzalutamide (2012), apalutamide (2018), darolutamide (2019)	Prostate
CD20	Rituximab (1997), ofatumumab (2009), obinutuzumab (2013)	NHL/CLL
HER-2	Trastuzumab (1998), pertuzumab (2012), T-DM1 (2013), neratinib (2017), tucatinib (2020), margetuximab (2020)	Breast
BCR-ABL	Imatinib (2001), dasatinib (2006), nilotinib (2007), bosutinib (2012), ponatinib (2012)	CML
EGFR	Erlotinib (2004), afatinib (2013), gefitinib (2015), osimertinib (2015), dacomitinib (2018), amivantamab (2021)	NSCLC
VEGFR/PDGFR/C-KIT	Sorafenib (2005), sunitinib (2006), pazopanib (2009), axitinib (2012), cabozantinib (2016), lenvatinib (2016), tivozanib (2021)	RCC
ALK	Crizotinib (2011), ceritinib (2014), alectinib (2015), brigatinib (2017), lorlatinib (2018)	NSCLC
PARP	Olaparib (2014), rucaparib (2016), niraparib (2019)	Ovary
CDK4/6	Palbociclib (2015), ribociclib (2017), abemaciclib (2017)	Breast
PD-1/PD-L1	Atezolumab (2016), nivolumab (2017), pembrolizumab (2017), durvalumab (2017), avelumab (2017)	Bladder

* First ever drug approved by FDA for the treatment of cancer. ^1^ MOPP: mechlorethamine, vincristine, procarbazine, prednisone; ^2^ ABVD: doxorubicin, bleomycin, vinblastine, dacarbazine; ^3^ PVB: cisplatin, vinblastine, bleomycin; ^4^ BEP: bleomycin, etoposide, cisplatin; ^5^ TIP: paclitaxel, ifosphamide, cisplatin; RCC: renal cell carcinoma (clear cell type); CML: chronic myelogenous leukemia; NSCLC: non-small-cell lung cancer; NHL/CLL: non-Hodgkin lymphoma/chronic lymphocytic leukemia.

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
