# Peer review of "Stem Cell Theory of Cancer: Implications for Drug Resistance and Chemosensitivity in Cancer Care"

_cancers, 2022, doi:10.3390/cancers14061548_

Round 1

Reviewer 1 Report

In this review,  Tu  et al. highlighted the role of cancer stem cell theory in cancer cell resistance,  and focused their attention on recent advance in the field. This reviewer has  these major comments:

- the  abstract is “copy and paste” of the same paragraph present in the introduction: the authors have to do more efforts to rewrite an original abstract;

- in the first paragraph of introduction  the authors used  a language that  is  divulgative  and not suitable for a scientific audience. This reviewer suggests to rewrite it in a more scientific way, supported by scientific literature and terms. This suggestion is also for other parts of the review;

-the quality of the figure is not good;

-the paragraph on “Stem cell origin” is not accurate in the description, something is described in dettails , something not: the authors have to decide better on what they want to focus on;

-this reviewer suggests to do another figure on the implication of stem cell hypothesis in cancer resistance, according to the conclusion :” A stem cell theory of cancer unites chemoresistance with chemosensitivity, connects  progenitor cells with progeny cells, and links multicellularity with the microenvironment. Importantly, it clarifies genetic content vs cellular context, delineates drug vs therapy development, enlightens precision medicine vs integrated medicine and targeted therapy vs  multimodal therapy, in patient care.”

Reviewer 2 Report

Thank you for giving me the opportunity to review this manuscript. For the following reasons, I do not consider the manuscript strong enough for publication in Cancers:

  • The topic is not new and there is no need for an additional publication about a summary of CSC, drug resistance, and chemosensitivity.
  • There is a lack of figures that would enhance the readability of the manuscript.
  • The authors cite their own work and are missing more important studies.

Author Response

Please, see attached file.

Reviewer 3 Report

this is an interesting paper highlighting stem cells and drug resistance in cancer.

While there is some interesting background information this paper would be of greater interest to the audience if more information about drug development and clinical trials were discussed (e.g in a diagram or table).

Author Response

Please, see attached file.

Round 2

Reviewer 1 Report

The revised version of the paper is ok

Author Response

The revised version of the paper is ok.

We thank the reviewer for your comments and support of this paper!

Reviewer 2 Report

I approve an improvement  of the manuscript. Unfortunately, the modifications do not improve the manuscript sufficiently.

Major weaknesses:

Table 1 summarizes the history of oncological drug approval, which can be found in specific manuals. Also, Figure 2 does not show great effort to improve the manuscript. The knowledge was already described 3-4 years ago.  

The criticized self-citation was neither commented nor corrected by citing the key studies.

The novelty why this review is needed is still missing. 

Author Response

We thank the reviewer for your insightful comments and suggestions. We have provided our responses below in bold with changes highlighted in red underneath the reviewer comments. We have incorporated all changes recommended by the reviewers with the changes highlighted in red in the manuscript.

I approve an improvement  of the manuscript. Unfortunately, the modifications do not improve the manuscript sufficiently.

Major weaknesses:

Table 1 summarizes the history of oncological drug approval, which can be found in specific manuals. Also, Figure 2 does not show great effort to improve the manuscript. The knowledge was already described 3-4 years ago.

Table 1 showed selected drug approvals and their indications as well as their relationship to drug development in the context of this manuscript. I will be very surprised if the reviewer can find the same information in this format in any specific manuals.

For example, we have used second-line ifosfamide and paclitaxel in TIP to treat drug-resistant testicular cancer (Table 1, row 2). However, drug development of ifosfamide and paclitaxel by itself has not improved overall survival from about 70% in the 1970s to over 90% today (ref 20). The idea that therapy development rather than drug development improves overall survival and overcomes drug resistance is unique (section 11). Importantly, advancement of therapy development in contrast to drug development is dependent on a proper cancer theory, which Figure 2 illustrates. Nevertheless, most people today still think in terms of drug development rather therapy development. To our knowledge, a unified theory of cancer based on a stem-cell origin of cancer is novel and has not yet been described in detail with respect to drug resistance like we do in this manuscript.

The criticized self-citation was neither commented nor corrected by citing the key studies.

It is true that the concept of cancer stem cells and drug resistance is well accepted. However, a stem cell theory that considers all cancer hallmarks, including metastasis, heterogeneity, dormancy, and immunity, besides drug resistance, in the origin and nature of cancer is still essential, and likely impactful, by galvanizing innovative ways to improve overall survival and overcome drug resistance.

We have deleted those (previous) references 2, 22, 32, and 40 that support this idea of a comprehensive stem cell origin of cancer, especially if the reviewer think that this unique perspective of a unified theory of cancer (as opposed to the idea of cancer stem cells) and drug resistance will become better accepted without need for additional supporting references.

The novelty why this review is needed is still missing.  

The theory of a stem cell origin of cancer is novel and unique (compared with the idea of cancer stem cells) because it alludes to a unified theory of cancer and suggests that cancer is a stem cell disease – considering both normal stem cells and cancer stem cells, connecting progenitor cells with progeny cells, and linking multicellularity to the microenvironment. In addition, it clarifies genetic content vs cellular context, delineates drug vs therapy development, enlightens precision medicine vs integrated medicine and targeted therapy vs multimodal therapy, which the idea of cancer stem cells by itself does not do.

Reviewer 3 Report

Changes made, including the addition of the table of anticancer agents, are appropriate.

Author Response

Changes made, including the addition of the table of anticancer agents, are appropriate.

We thank the reviewer for your insights and approval of this manuscript!